# Photoacoustic Image Analysis of Dental Tissue Using Two Wavelengths: A Comparative Study

Marco P. Colín-García [1], Misael Ruiz-Veloz [2], Luis Polo-Parada [3], Rosalba Castañeda-Guzmán [1], Gerardo Gutiérrez-Juárez [2], Argelia Pérez-Pacheco [4],* and Roberto G. Ramírez-Chavarría [5],*

1 Instituto de Ciencias Aplicadas y Tecnología, Universidad Nacional Autónoma de México, Mexico City 04510, Mexico; dreizehn@ciencias.unam.mx (M.P.C.-G.); rosalba.castaneda@icat.unam.mx (R.C.-G.)
2 División de Ciencias e Ingenierías, Universidad de Guanajuato, Guanajuato 37150, Mexico; ruizvg2012@licifug.ugto.mx (M.R.-V.); ggutj@fisica.ugto.mx (G.G.-J.)
3 Department of Medical Pharmacology and Physiology, Medical School, The University of Missouri, Columbia, MO 65212, USA; poloparadal@missouri.edu
4 Unidad de Investigación y Desarrollo Tecnológico, Hospital General de México "Dr. Eduardo Liceaga", Mexico City 06726, Mexico
5 Instituto de Ingeniería, Universidad Nacional Autónoma de México, Mexico City 04510, Mexico
* Correspondence: argeliapp@ciencias.unam.mx (A.P.-P.); rramirezc@iingen.unam.mx (R.G.R.-C.)

**Abstract:** This work compares photoacoustic images of a tooth by analyzing the signals generated with wavelengths 532 and 355 nm. This comparison addresses the differences in the optical properties of dental tissue for these wavelengths that affect the resulting photoacoustic images. A pulsed Nd:YAG laser was used to illuminate a complete extracted tooth sample, and 2D photoacoustic images (PAIs) were reconstructed using the single-sensor scanning synthetic aperture focusing technique (SSC-SAFT), which is a suitable method for our experimental system with forward detection mode. Signal comparison was conducted using sinogram, signal-to-noise ratio (SNR), root mean square (RMS), arrival time, maximum amplitude, and fast Fourier transform (FFT). PAI comparison utilized intensity profile, edge correlation, and image composition tools. The signal analysis revealed that at 532 nm, the signals exhibited longer decay time and a wider distribution of vibration frequencies due to higher laser pulse energy and greater optical penetration depth. Conversely, at 355 nm, the signals had shorter decay times and a lower frequency distribution, which was attributed to lower energy but improved optical absorption, resulting in reconstructed images with better sharpness and contour definition. This study contributes to the advancement of photoacoustic imaging technology in dentistry by providing insights that could optimize signal generation and image reconstruction for dental tissue.

**Keywords:** photoacoustic effect; photoacoustic signal; photoacoustic imaging; dental hard tissue; ultrasound transducer

## 1. Introduction

Photoacoustic tomography (PAT) emerges as a medical and biophotonic imaging technique, fusing optical and acoustic principles to produce detailed photoacoustic images (PAIs) of biological tissues. Currently, it is possible to obtain a PAI of any part of the human body, including both soft and hard tissues. PAT leverages the photoacoustic effect by converting absorbed optical energy into acoustic energy, generating acoustic waves through the optical absorption of the illuminated material [1,2].

PAT is a non-invasive imaging modality that combines high optical contrast and penetration depth with the high spatial resolution of ultrasound in tissue [3]. Its particular strength, relative to pure optical imaging modalities, lies in the detection of ultrasound rather than light, which reduces scattering and attenuation in tissue [4]. PAT can achieve

an imaging depth of several centimeters, whereas techniques such as optical coherence tomography, two-photon microscopy, confocal microscopy, or Raman imaging are limited to a few millimeters in depth [5]. The high contrasts and biosafety of PAT provide natural advantages in mapping the physiological structure of biological tissues, including breast cancer detection, brain imaging, vascular disease monitoring, and joint imaging [6]. Furthermore, PAT uses non-ionizing radiation, which is safer than the ionizing radiation employed in techniques like X-rays, thereby reducing the risks associated with prolonged exposure during clinical diagnosis.

To generate acoustic waves from light, a short-pulse laser can be used as an excitation source, with a wavelength and optical power that depend on the optical properties of the target sample. This illumination heats the area, causing an increase in temperature and initial pressure $p_0$, which leads to the generation of acoustic waves, generally in the MHz range. Detecting these acoustic waves with an appropriate ultrasound transducer is known as capturing a photoacoustic signal [7].

Depending on the arrangement of the transducers and the mode of detection of the acoustic waves (backward, forward, or lateral), different reconstruction algorithms can be implemented to generate an image from the acquired signals. In image reconstruction, two fundamental problems can be distinguished: the direct problem and the inverse problem. The direct problem involves generating signals from the incidence of a pulsed laser on an absorbent material (signal acquisition). In contrast, the inverse problem involves determining the origin of the signals (image reconstruction). Some reconstruction algorithms used in PAT to solve the inverse problem are time-reversal [8,9], delay-and-sum [10,11], fast Fourier transform [12,13], and back-projection [14,15]. These algorithms may or may not consider the photoacoustic wave equation.

Currently, few studies of hard biological tissue have been performed with PAT. In medical terms, hard tissue refers to dense and resistant tissues in the human body, such as bones and teeth. Particularly, the crown of a tooth is composed of three layers of tissue: enamel (outer layer), dentin (middle layer), and dental pulp (inner layer). Enamel and dentin differ in structure but have similar components. The primary biological material in enamel and dentin is carbonated hydroxyapatite (85% in enamel, 47% in dentin), followed by water (12% in enamel, 20% in dentin) and proteins and lipids (3% in enamel, 33% in dentin) [16].

Optical absorption is an important property to consider because efficient ultrasound generation depends on the absorption characteristics of the dental tissue in the wavelength band emitted by the pulsed laser [17]. In the bands around 532 and 355 nm, hydroxyapatite exhibits stronger absorption than water [18]. Additionally, dentin shows greater absorption coefficients $\mu_a$ than enamel in these bands (dentin: $\mu_{532} \approx 0.55$ and $\mu_{355} \approx 0.85$; enamel: $\mu_{532} \approx 0.35$ and $\mu_{355} \approx 0.55$) [19]. Considering that optical penetration depth is inversely proportional to the absorption coefficient ($1/\mu_a$), the 532 nm wavelength penetrates deeper into the dentin and enamel layers.

Some of the laser applications in dentistry have been implemented for cavity preparation, caries prevention, cleaning, and dental surgery. Laser radiation enables controlled ablation and vaporization processes, favorable hemostasis, and sterilization [20]. Laser-generated ultrasound in photoacoustics has the potential to complement conventional radiography as an imaging technique in clinical dentistry [21].

One of the first applications of ultrasound in dental imaging involved obtaining a 2D contour image of the external enamel surface and the enamel–dentin junction by scanning around an extracted human tooth [22]. Subsequent studies focused on measuring enamel thickness and periodontal pocket depth [23]. Later, the use of double-contrast photoacoustic tomography was explored for detecting early dental lesions. In this method, one contrast is related to optical absorption and the morphological and macrostructural characteristics of the teeth, while the other contrast is associated with the microstructural and mechanical properties of the tissue [6]. PAI applications have also been implemented on enamel to identify microscopic cracks, lesions, and the distribution of dental calcium

for demineralization studies [24]. Finally, PAI has been used to non-invasively monitor gingival health in humans by quantifying periodontal pocket depth, offering advantages over the manual periodontal probing method [25,26].

In this work, we studied the acquired photoacoustic signals and the reconstructed 2D PAIs of a cross-section of a dental tissue sample using two wavelengths generated by a pulsed Nd:YAG laser (532 and 355 nm). We used the single-sensor scanning synthetic aperture focusing technique (SSC-SAFT), which is an image reconstruction method [27] designed for a circular scan using a single acoustic sensor and a system that allows the sample to be illuminated and rotated around a fixed axis. This algorithm is notable because it considers the sensor size, eliminates unwanted artifacts, generates a more distinguishable image, and consumes less time and computational resources than conventional algorithms. Given the optical properties of dental tissue, 532 nm has a greater optical penetration depth into enamel and dentin than 355 nm. Will this produce differences in the image reconstructed by photoacoustic imaging? Our preliminary study provides insights to answer this question and contributes to the field of photoacoustic imaging for dental evaluations.

## 2. Materials and Methods

The experimental system used in this work is shown in Figure 1. The equipment consisted of (1) a Quantel Brilliant b pulsed Nd:YAG laser with second harmonic (532 nm, 9.5 mJ pulse energy) and third harmonic (355 nm, 1.9 mJ pulse energy), 5 ns pulse width, and 10 Hz pulse repetition rate; (2) a digital oscilloscope Tektronix DPO5204B with 2 GHz bandwidth and 10 GS/s sampling rate; (3) an Olympus Immersion Transducer A326S-SU with 4.92 MHz center frequency, 2.57 MHz bandwidth, and 9.52 mm element diameter; (4) a Thorlabs PDA10A Silicon Amplified Detector; (5) a Nema 17 stepper motor with 1.8° step angle; (6) Arduino UNO board; (7) a laptop with AMD Ryzen 5, 2.38 GHz, 12GB RAM; (8) a beam splitter; a (9) plano-convex lens with 40 cm focal length; (10) an acrylic container with dimensions L 20 × W 10 × H 15 cm; and (11) a dental sample with a crown diameter of 1.2 cm. The study sample consisted of a third molar extracted from a patient experiencing pain. The tooth exhibited no visible damage to the sides of the crown.

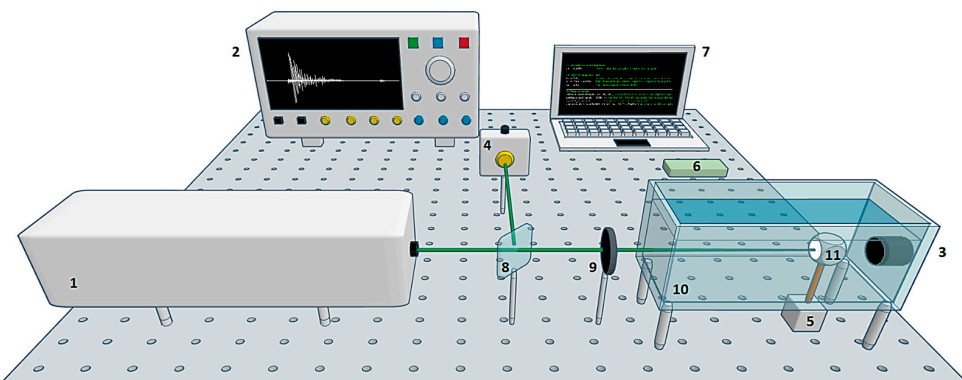

**Figure 1.** Experimental system scheme. The laser was focused on the surface of the sample to generate acoustic waves. These waves were detected in forward mode by an ultrasound transducer. The signal was recorded on the computer. A stepper motor was used to rotate the sample by 1.8°. The cycle was repeated until completing one turn: (**1**) pulsed laser, (**2**) oscilloscope, (**3**) ultrasound transducer, (**4**) photodetector, (**5**) stepper motor, (**6**) Arduino board, (**7**) laptop, (**8**) beam splitter, (**9**) lens, (**10**) container, and (**11**) sample.

### 2.1. Signal Acquisition

The experimental system developed for the acquisition of photoacoustic signals consisted of a single sensor in forward detection mode. The amplitude of the acquired signals $V_{out}$ depends mainly on the optical fluence $\Phi$ deposited in the sample, the optical absorption coefficient at a certain wavelength $\mu_a$, and the Grüneisen parameter $\Gamma$ (the constant of

proportionality between the absorbed light and the initial pressure). This relationship is expressed as follows:

$$V_{out} \propto \Phi \mu_a \Gamma. \tag{1}$$

The experimental conditions were kept the same for the subsequent comparative analysis, except for the laser pulse energy parameter, as the amplitudes of the signals were adjusted to be similar for both wavelengths. The laboratory temperature was maintained at a constant 20 °C. The tooth was immersed in a container with water, as shown in Figure 2. Both the transducer and laser path were kept fixed. Signal acquisition was executed with a MATLAB code to automate the equipment shown in Figure 1. The sample was scanned at 532 nm and later at 355 nm.

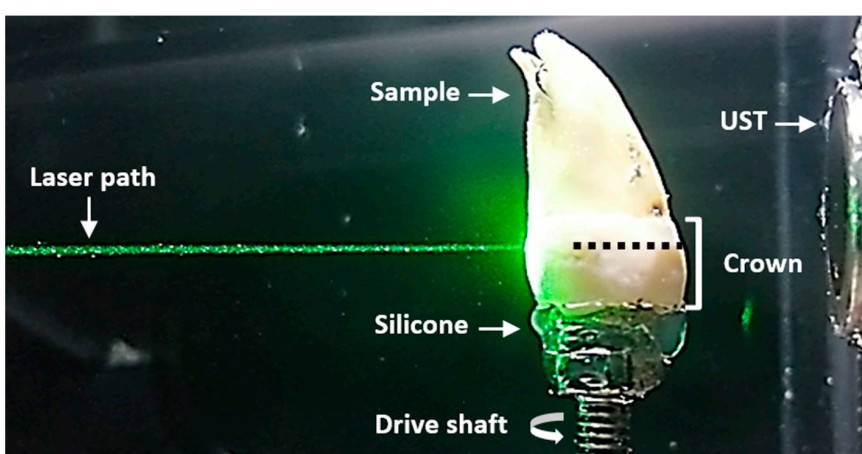

**Figure 2.** Sample scanning plane. Lateral view of the sample immersed in water. The laser path and ultrasound transducer (UST) remained fixed for each sample rotation. The plane where the laser was focused is indicated by a dotted black line.

Initially, a train of focused laser pulses was sent to the crown of the sample. Each signal (averaged over 30 pulses) was displayed on the oscilloscope, and the data were transferred to the laptop to generate a data file (.txt). The stepper motor then rotated the sample with a step angle of 1.8° to complete one cycle. This process was repeated until 200 cycles were completed, rotating the sample a full 360°. Data files were stored with 10,000 samples in a 50 µs window (200 MS/s sampling rate). The sample was rotated in the same scanning plane using a single transducer to simulate a circular array of 200 physical sensors placed around the sample. The time taken to acquire the 200 signals was approximately 35 min.

### 2.2. Image Reconstruction Algorithm

The SSC-SAFT method [27] is suitable for an experimental setup involving a circular scan using a single acoustic sensor and a system that rotates the sample to emulate a circular sensing surface. Due to its symmetry, this method is invariant to the backward or forward detection mode. The difference in the detection mode is the distance between the acoustic source and the transducer. In the forward mode, the acoustic waves generated travel a greater distance since they propagate through the sample and then in the water, necessitating an average value for the speed of sound in different media to be estimated.

This method incorporates modifications of the SAFT method [28] to reconstruct an image, focusing exclusively on the sensor detection region. It performs a segmented reconstruction, where each segment is delimited by the size, position, and detection area of the acoustic sensor within the image grid. To implement the algorithm, a computational grid (space where the calculations are performed) is initially defined. Since the image is a matrix where each element represents the position of a pixel, and its numerical entry represents the intensity value, the objective is to calculate this numerical entry for each

array element based on the electrical signals acquired during the scanning. The result is an intensity image representing the initial pressure distribution $p_0$ that generates these signals.

Figure 3 presents the SSC-SAFT process for obtaining images in the case of circular scanning with eight sensor positions. In Figure 3A, the red dotted circle represents the path covered by the sensor during scanning, and the red hexagon represents the position of the hypothetical source to be reconstructed. Once the computational grid of $Nx \times Nx = 13 \times 13$ entries is constructed, the positions of the sensors $s_i$ are located during scanning. The reconstruction is then performed over the line of pixels $l_1^j$ that connects $s_1$ with the position of the diametrically opposite pixel $j$. This process is repeated for every position $s_i$, obtaining an image of $N$ lines $l_i^j$. Since the reconstruction is performed in a plane, the resulting image can be understood as an image of a slice of the sample. The portion of the sensor involved in this plane is the length $A$ of this segment, and the number of pixels it covers is $A/\Delta x$. Considering that the index $k$ describes the pixel coordinates that constitute the effective detection region $s_{i,k}$, a segmented reconstruction region is generated that covers the portion of the sample that was illuminated and its corresponding detection surface. Figure 3B presents the last part of the reconstruction process. In this example, the sensor $s_{1,k}$ has a detection region equivalent to three pixels in length $k = 1, 2, 3$. For each sensor position $s_{i,k}$, the reconstruction is repeated on the neighboring line $l_{i,k}^j$ so that the effective detection area of the sensor is covered in the grid. The grayscale value $I_{SSC-SAFT}$ of a pixel in a photoacoustic image obtained with the SSC-SAFT reconstruction method is summarized in the following relationship:

$$I_{SSC-SAFT}\left(l_{i,k}^j\right) = V\left[s_i, t - \tau\left(s_{i,k}, l_{i,k}^j\right)\right]_{t=0}, \tag{2}$$

where $j = 1, 2, \ldots, M$ and $k = 1, 2, \ldots, round(A/\Delta x)$, $V$ is the voltage vector acquired during a time $t$ by the sensor $s_i$, and $k$ is the position of a pixel that conforms to the effective detection region with length $A$. $l_{i,k}^j$ indicates the pixel $j$ in the pixel line $l_{i,k}$, formed by $M$ pixels, connecting $k$ with its diametrically opposite pixel; $\tau$ is the time delay for a given medium-speed sound.

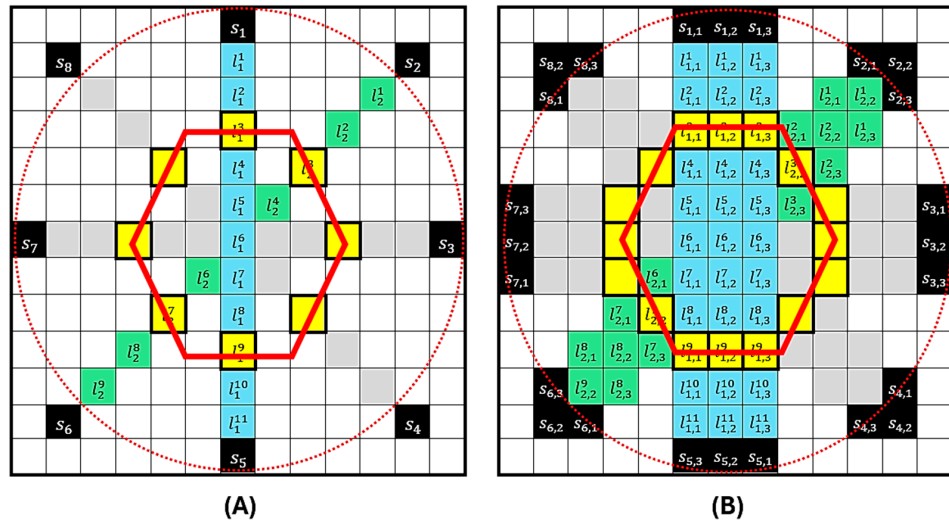

**Figure 3.** Representation of the SSC-SAFT process for the case of a circular scan with eight sensor positions. In (**A**), the red circle represents the sensor's path during scanning, and the red hexagon represents the position of the source to be reconstructed. The computational grid of $13 \times 13$ entries shows the positions of the sensors $s_i$. The reconstruction is performed on the pixel line $l_1^j$. The process is repeated for each position $s_i$. In (**B**), the sensor $s_{1,k}$ has a detection region equivalent to three pixels in length $k = 1, 2, 3$. For each sensor position $s_{i,k}$ the reconstruction is repeated on the neighboring line $l_{i,k}^j$.

The SSC-SAFT method is independent of any mathematical model, such as the photoacoustic wave equation, and is suitable for recovering the image of any sample that generates an acoustic signal.

In this work, the experimentally acquired signals with the system shown in Figure 1 were used for 2D PAI reconstruction through a code developed in MATLAB R2021a. The parameters considered to generate the image were as follows: computational grid size of $Nx \times Nx = 2048 \times 2048$, radius of the circular array of sensors $r = 0.158$ m, length of the sensor $A = 0.00952$ m, and the estimated average value of the speed of sound in the different media (tooth and water) $c = 1350$ m/s. The computational calculation time to reconstruct the image with this resolution and using the complete signals was approximately 81 s.

## 3. Results

Figure 4 shows the 2D PAIs of the sample obtained through SSC-SAFT at (A') 532 and (B') 355 nm. Additionally, (C') shows the optical image of the same scanning plane, and (D') depicts the X-ray image of the sample.

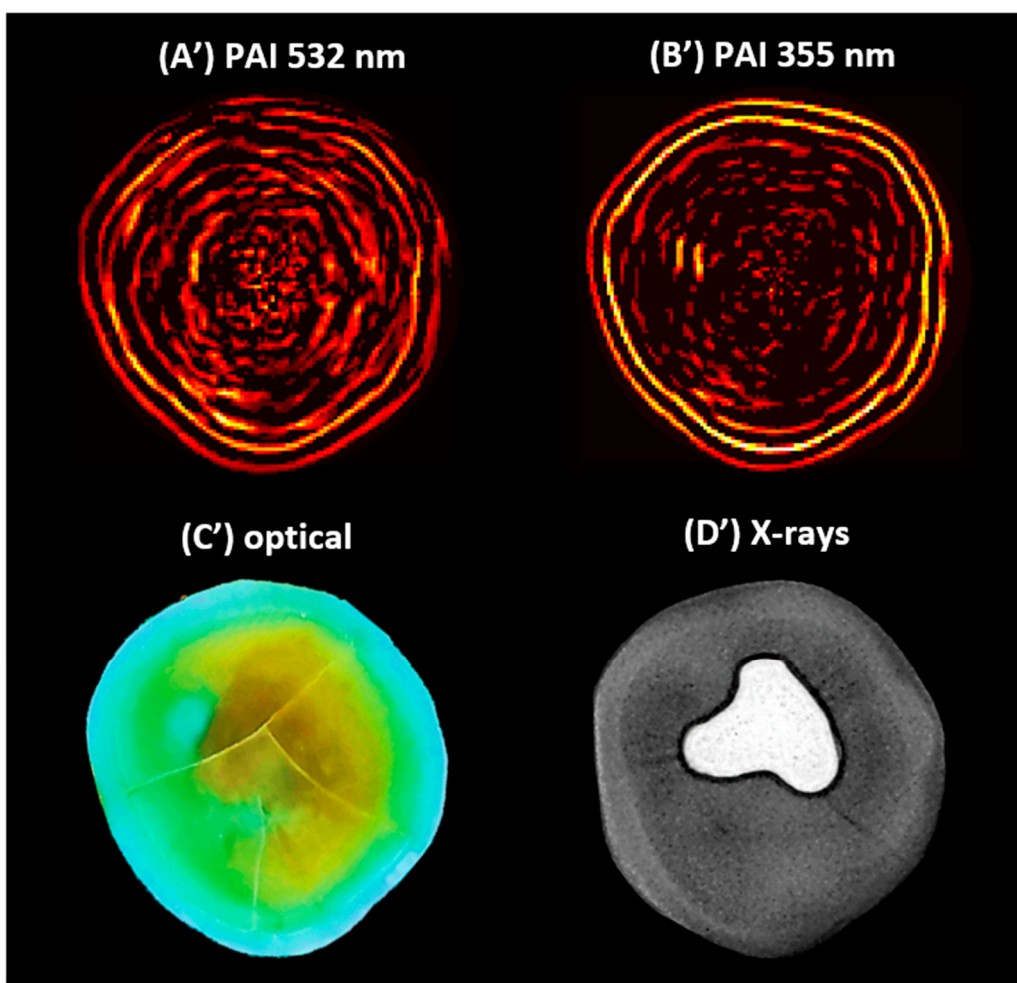

**Figure 4.** The 2D photoacoustic images (PAI) of a dental sample (**A'**) using 532 nm, (**B'**) using 355 nm, (**C'**) optical image of the scanning plane, and (**D'**) X-ray image of the sample, which shows an amalgam on the occlusal aspect of the crown.

The results show that the set of signals acquired at 532 and 355 nm differ in decay time and waveform, while the images reconstructed with SSC-SAFT differ in intensity and edge definition. To quantitatively compare these results, signal and image processing was performed using MATLAB tools. For the analysis, the complete signals were considered. Figure 5 presents the comparative analysis of the acquired signals and reconstructed images.

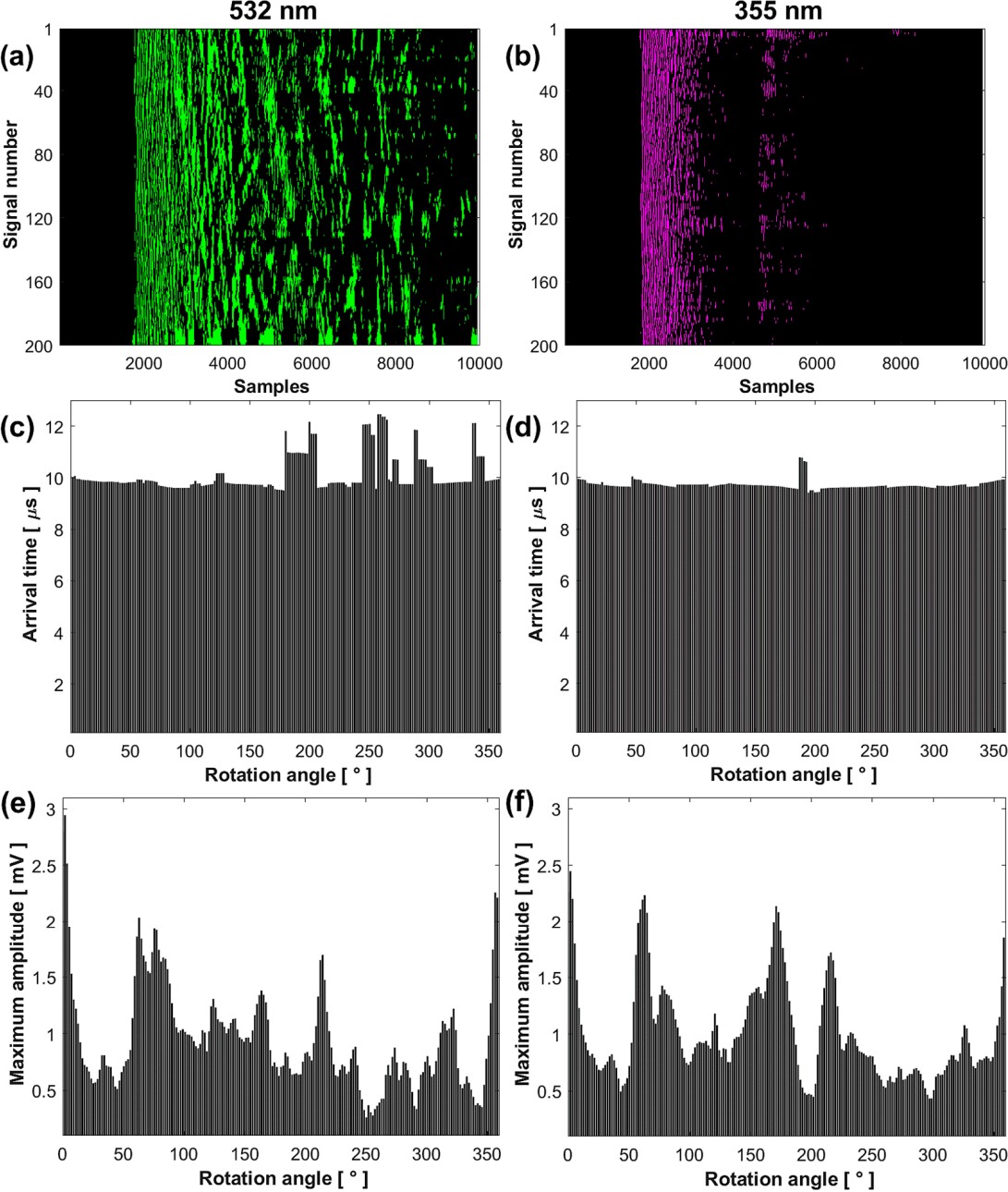

**Figure 5.** *Cont.*

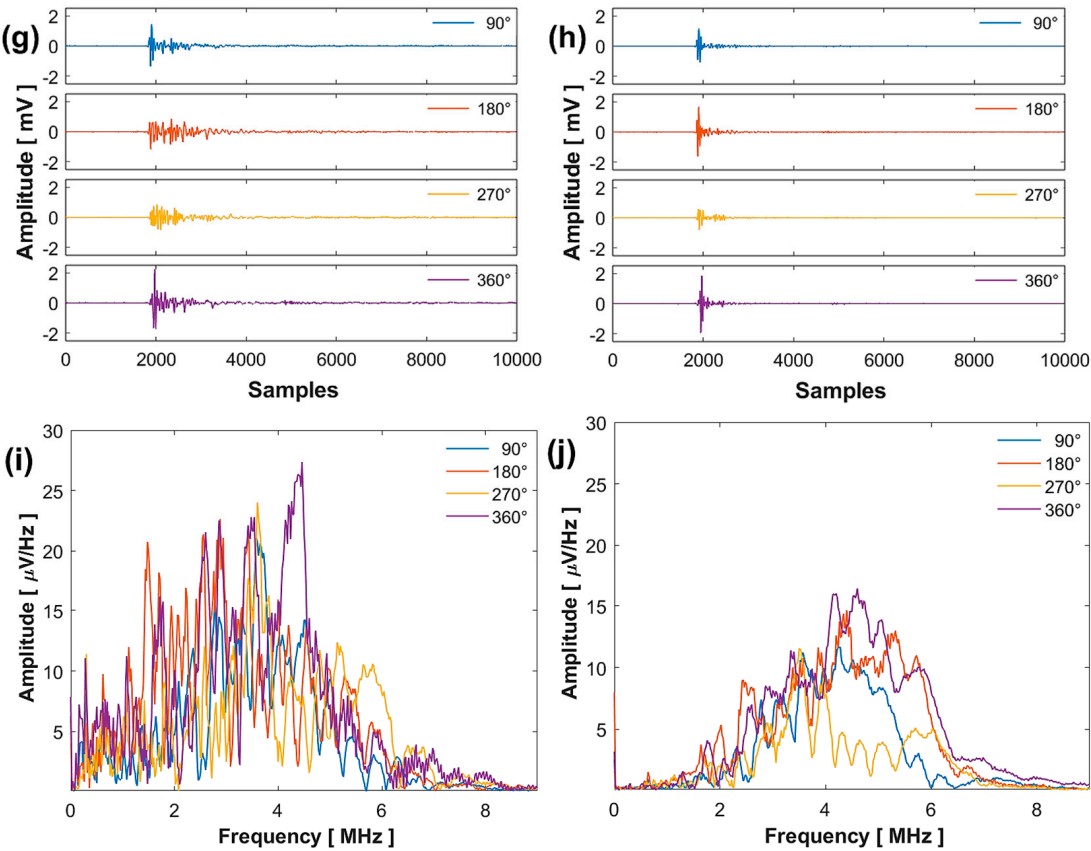

**Figure 5.** Comparative analysis of the results obtained at 532 (left column) and 355 nm (right column): sinogram (**a**,**b**), arrival time (**c**,**d**), maximum amplitude (**e**,**f**), signals (**g**,**h**), and FFT (**i**,**j**).

### 3.1. Photoacoustic Signal Analysis

Figure 5a,b show that the sinogram for the signals obtained at 532 nm has a longer decay time, presenting a wide band of intensities from sample number $\approx 1900$ to 10,000. In contrast, the signals obtained at 355 nm present two bands: the first from sample number $\approx 1900$ to 3900 and the second from $\approx 4700$ to 5900.

The average SNR value of the signals at 532 nm was $4.3537 \pm 2.3378$ dB, while at 355 nm, it was $7.6208 \pm 2.1001$ dB. This higher SNR value at 355 nm indicates that the conversion of light into sound is more efficient, despite the lower laser pulse energy. The average RMS value of the signals at 532 nm was $0.1038 \pm 0.0243$ mV$_{RMS}$ and at 355 nm was $0.0692 \pm 0.0216$ mV$_{RMS}$, indicating that the signals at 532 nm have a higher energy content. The high standard deviation in the average SNR and RMS values is due to the sample's non-homogeneity in the scanned plane, making the system sensitive to the irregularities present in the sample.

Figure 5c,d show that the signal arrival times are similar from the rotation angle $0°$ to $178.2°$ for both wavelengths. This indicates that the acoustic source generated by optical absorption originates from approximately the same area of the sample, very close to its surface. Differences in arrival times for the remaining angles are due to irregularities in the illuminated area, increasing the travel distance of the acoustic waves to the transducer. The average arrival time value of the signals from angle $0°$ to $178.2°$ at 532 nm was $9.7856 \pm 0.1379$ µs, while at 355 nm, it was $9.7300 \pm 0.0081$ µs.

Figure 5e,f show that the maximum amplitude of the first peaks of the signals depends on the optical absorption of the illuminated region of the sample. The graphs illustrate continuous variations in signal amplitude depending on the rotation angle, with maximums and minimums associated with different optical absorption capacities. These

amplitudes differ at each measurement point due to the non-homogeneous distribution of hydroxyapatite, the main absorber, around the sample.

Figure 5g,h present four signals acquired at four different rotation angles, with their respective FFTs shown in Figure 5i,j. The spectrum at 532 nm displays a larger distribution of frequencies with greater amplitude compared to 355 nm, considering that the signals contain the vibration frequencies of the complete system.

### 3.2. Photoacoustic Image Analysis

In Figure 6a,b, for a comparative analysis of the sample interfaces at both wavelengths, the intensity profile was taken along a line directed toward the center of the PAIs. These profiles are represented by a dotted white line. The first peak of the profile (A = pixel 30) associated with the external contour of the sample (water–enamel interface) is at the same distance for both wavelengths. It is possible that the third peak (B = pixel 39) is associated with the next interface inside the sample (enamel–dentin junction). The rest of the peaks could be associated with acoustic reflections produced in the system.

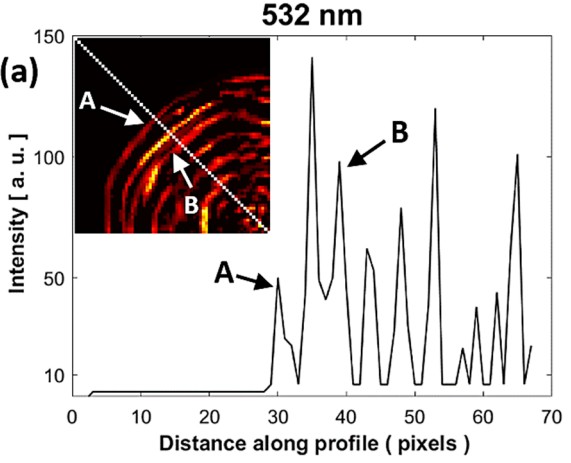 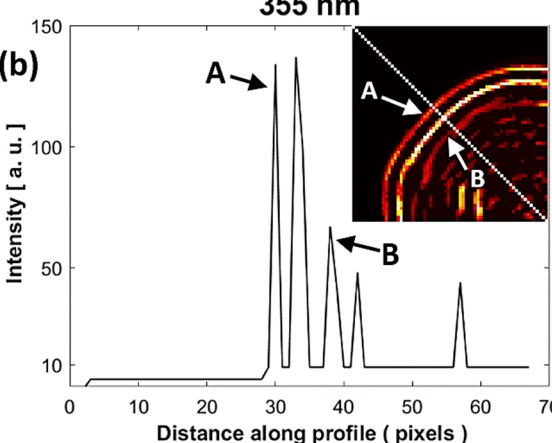

**Figure 6.** Comparison of the intensity profile of the PAIs at (**a**) 532 nm and (**b**) 355 nm. The intensity profile was taken along a white dotted line of the PAIs shown in subfigures of (**a**,**b**). A: water-enamel interface, B: enamel-dentin junction.

As shown in Figure 7, a final analysis was performed to compare the PAIs using image composition. The purpose of this analysis is to identify the similarities and differences by superimposing the PAIs obtained with the two wavelengths and the optical image of the physical sample. To highlight the salient features of the images, they were binarized. Figure 7 shows (a') the optical image of the sample cut on the same scanning plane, (b') the PAI obtained at 532 nm, (c') the PAI obtained at 355 nm, (d') the binarized image of the optical image, (e') the binarized PAI at 532 nm, (f') the binarized PAI at 355 nm, (g') the binarized image composed of PAIs obtained at 532 and 355 nm, (h') the binarized image composed of the optical image and PAI at 532 nm, and (i') the binarized image composed of the optical image and PAI at 355 nm.

The correlation of the binarized PAIs shown in Figure 7e',f', considering the first outer edge (I), was 0.7399. The correlation for the two outer edges (I and II) was 0.6296; for the three outer edges (I, II, and III), it was 0.5444; and for the entire images, it was 0.4093. In Figure 7g'–i', the gray pixels show the areas where the two images are similar, while the magenta and green pixels show the areas where the respective images differ. In Figure 7g', it is observed that the binarized PAIs are very similar in the external contours, but within the edges, there are pixels with greater intensity in the PAI at 532 nm.

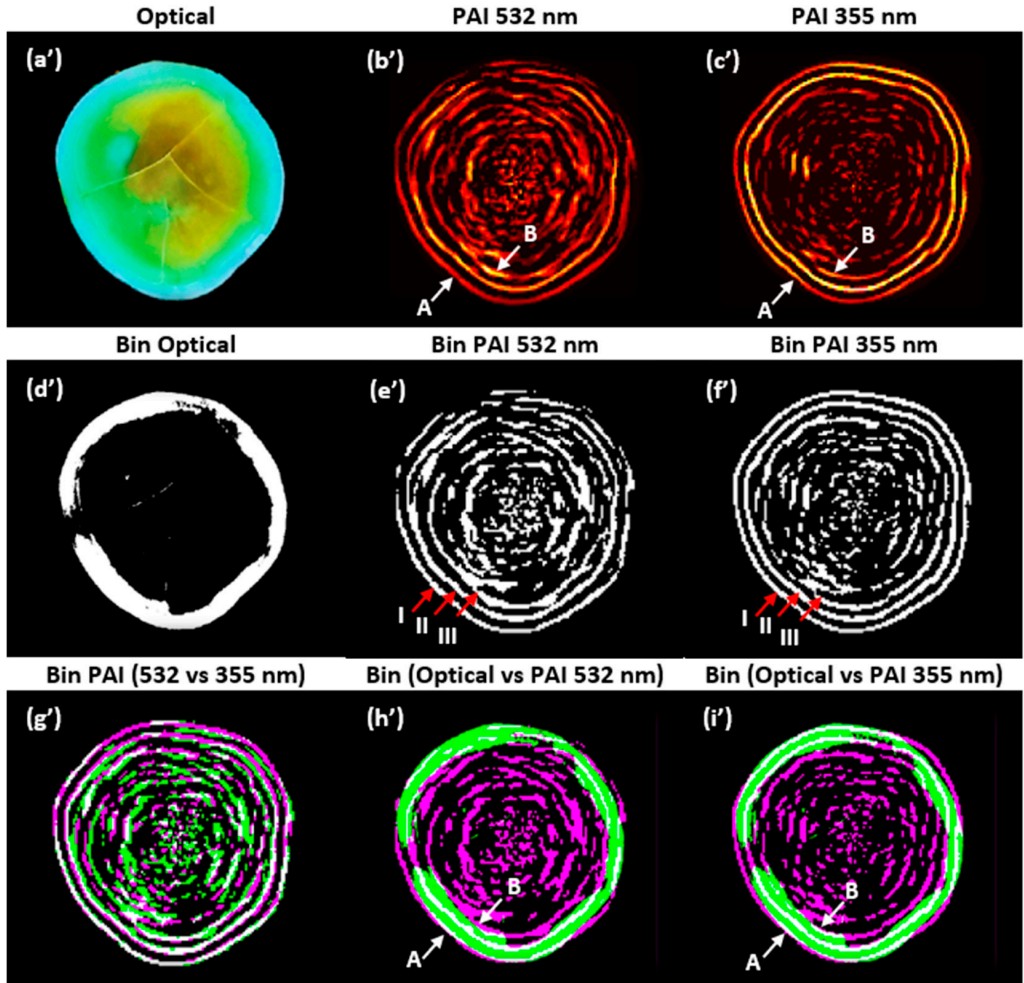

**Figure 7.** Composite images: non-binarized (**a'**–**c'**); binarized (**d'**–**i'**). PAIs were reconstructed with the SSC-SAFT algorithm. A: water–enamel interface, B: enamel–dentin junction. I, II, and III: outer edges.

In Figure 7h',i', it is observed that the external contour of the binarized optical image closely matches the shape of the external contour of the binarized PAI (water–enamel interface). However, the similarity is not perfect due to inaccuracies in cutting the sample on the same scanning plane with the tools used, as well as the error introduced by the angle of the photograph taken of the sample. The internal contour of the optical image (enamel–dentin junction) aligns with the third edge identified in the intensity profile analysis (B = pixel 39). In future work, we will focus on verifying the thickness of the enamel layer (A–B) using our system.

## 4. Discussion

Our experimental system allowed for the detection of acoustic waves generated by the optical absorption of the dental sample at 532 and 355 nm without the need for signal amplification and filtering. To achieve a signal with an amplitude of approximately 2 mV, we used a pulse energy of 9.5 mJ at 532 nm and 1.9 mJ at 355 nm, which is five times less energy. At the end of the measurements with these energies, microscopic examination showed no visual damage to the sample's surface.

Considering the optical absorption coefficient values from reference [19], we agree that the sample components have greater absorption at 355 nm. Therefore, the interaction of the 532 nm laser with the sample was higher, as it penetrated more deeply into the dentin and enamel layers. The signals acquired at 532 nm had higher energy content and a longer decay time compared to the signals acquired at 355 nm (see Figure 5a,b). The fre-

quency spectrum at 532 nm showed a greater frequency distribution and amplitudes than at 355 nm. These differences imply that the system has more energy, resulting in the appearance of vibration frequencies that are not evident at lower energy levels.

Furthermore, at 532 nm, the acoustic source was larger due to the greater illuminated volume along the path of light penetrating the enamel and dentin. A higher number of frequencies could interfere and produce undesirable results in the reconstructed image. Additionally, factors at each measurement point can cause variations in amplitude, arrival time, and signal shape. For example, irregularities in the sample's anatomy, the organic composition of the illuminated area, and the presence of dental stains affect the path of light, as well as the thickness of the layers through which the acoustic wave travels.

The average arrival time value calculated in the signal analysis reveals that the acoustic source generated from optical absorption originates at approximately the same distance for both wavelengths. This is verified through image analysis, first with the intensity profile in Figure 6a,b, and then with the composition of the PAIs in Figure 7g′, where the external contours associated with the water–enamel interface coincide for both wavelengths. Additionally, in Figure 7h′,i′, the external contour approximates the shape of the contour in the optical image.

Based on the intensity profile results, the third contour of the PAIs also approximates the contour of the enamel–dentin junction in the optical image, helping to verify that PAIs can approximate the thickness of the enamel. The advantage of implementing the SSC-SAFT algorithm is that it accounts for the signals acquired step by step using a single sensor. Moreover, the computational calculation time for image reconstruction with a resolution of 2048 by 2048 points using SSC-SAFT was approximately 11 s, compared to the time-reversal algorithm of the MATLAB k-Wave toolbox, which took approximately 2340 s.

In summary, one of the advantages of using 532 nm is that the signal decay time is longer, which could provide information about the structure or composition of the sample interior with other types of analysis. On the other hand, using 355 nm requires less laser pulse energy to achieve the same signal amplitudes as 532 nm. Since enamel absorbs 355 nm more efficiently, the penetration depth is lower. Therefore, the acoustic source is generated closer to the sample surface, resulting in PAIs with better definition of the external contours, good spatial resolution and contrast, and fewer artifacts.

## 5. Conclusions

In this work, a single-sensor system was implemented to acquire signals in forward detection mode from a dental hard tissue sample in vitro and generate its 2D image through photoacoustic tomography (PAT). The contribution of this work focuses on answering the following research question: Are there differences in the PAIs when illuminating a dental tissue sample with two wavelengths of an Nd:YAG laser, 532 nm and 355 nm? In response to this question, comparative quantitative analysis revealed both differences and similarities in the acquired signals and the reconstructed images.

In the signal analysis, notable differences were found in decay times, waveform, and frequency number. In the image analysis, differences were observed in pixel intensity and edge definition for both wavelengths. The use of 532 nm allowed for the acquisition of signals with a longer decay time, which could enable the study of the internal structure of the tooth in a non-ionizing manner but required a laser pulse energy five times higher compared to 355 nm. Conversely, the use of 355 nm allowed for signal acquisition with less energy, as the organic components on the surface of the dental tissue, mainly hydroxyapatite in the enamel, favor optical absorption. The images obtained with both wavelengths are very similar, although those obtained at 355 nm have a greater definition on the external edges.

The implementation of the SSC-SAFT reconstruction algorithm, which is fast and suitable for our signal acquisition system, where the sample is rotated, returned images with good spatial resolution, distinguishable shapes, minimal unwanted artifacts, good contrast differences, and sharpness. These preliminary results show that our system can

recover the contour of the tooth, identify areas with greater optical absorption around the sample, and approximate the thickness of the enamel. The results of this work could be significant in the development of imaging systems that complement optical techniques focused on the creation and design of dental implants or in studies focused on whitening treatments and monitoring of dental demineralization.

In conclusion, this study contributes to the search for non-ionizing imaging techniques and provides information to select a wavelength for use in dental research. It also supports the future implementation of photoacoustic imaging systems for dental tissue.

**Author Contributions:** Conceptualization, M.P.C.-G. and A.P.-P.; methodology, M.P.C.-G. and A.P.-P.; software, M.P.C.-G. and M.R.-V.; validation, R.C.-G., L.P.-P., G.G.-J. and R.G.R.-C.; formal analysis, M.P.C.-G., A.P.-P. and R.G.R.-C.; resources, A.P.-P. and R.G.R.-C.; writing—original draft preparation, M.P.C.-G., A.P.-P. and R.G.R.-C.; writing—review and editing, R.C.-G., L.P.-P., G.G.-J. and M.R.-V.; supervision, R.C.-G., A.P.-P. and R.G.R.-C.; funding acquisition, R.G.R.-C. All authors have read and agreed to the published version of the manuscript.

**Funding:** This work was supported by the grant UNAM-DGAPA-PAPIIT TA101423.

**Institutional Review Board Statement:** Not applicable.

**Informed Consent Statement:** Not applicable.

**Data Availability Statement:** Data is available if required.

**Acknowledgments:** We express our gratitude to Claudia Bravo Flores of the Servicio de Estomatología of the Hospital General de Mexico "Eduardo Liceaga" and to Gonzalo Montoya Ayala of the División de Estudios de Posgrado e Investigación of the Facultad de Odontología of the UNAM, for providing us with the dental samples to carry out this work. Marco P. Colín-García acknowledges CONAHCYT for their PhD study grant (CVU: 1085013).

**Conflicts of Interest:** The authors declare no conflicts of interest.

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
