# Peer review of "Photoacoustic Image Analysis of Dental Tissue Using Two Wavelengths: A Comparative Study"

_photonics, doi:10.3390/photonics11070678_

Round 1

Reviewer 1 Report

Comments and Suggestions for Authors

The article studied the acquired photoacoustic signals and the reconstructed 2D PAIs of a cross-section of a dental tissue sample using two wavelengths generated by a pulsed Nd:YAG laser.  This study contributes to the advancement of photoacoustic imaging technology in dentistry by providing insights that could optimize signal generation and image reconstruction for dental tissue.

There are some minor problems, which must be solved before it is considered for publication. I would like to present the following comments:

1.       The manuscript needs careful editing. The authors should particular attention to the format and resolution of multiple sets of images.

2.       In Line 249, the authors listed the average SNR value of the signals at 532 nm (4.3537) and 355 nm (7.6208), therefore the conversion of light into sound is more efficient at 355 nm. Make sure is there any mistake.

3.       In Line 250, can the authors describe how to solve the sensitivity of the system to irregularities in the sample or reduce the impact?

4.       When placing multiple figures together, the authors should pay attention to clarity. For example, the authors may refine figures 5(g) and 5(h), such as the alignment of the four labels, and the small font size of the axis scale. The color of the x-axis should be black in 5(i) and 5(j).

5.       Figure 5(i), 5(j), and Figure 6, the y-axis parameter lacks the corresponding unit.

Comments on the Quality of English Language

minor editing

Reviewer 2 Report

Comments and Suggestions for Authors

In the manuscript, a single-sensor photoacoustic system was constructed and used to acquire signals from a dental hard tissue sample and generate 2D image of its interior.

The authors show that the use of the laser with 355 nm wavelength allows one to obtain better definition of the external edges of the sample (the enamel-dentin interface), whereas 532 nm wavelength is better for studying deeper parts of the sample.

However, in both cases accurate reconstruction of the internal structure was achieved for near-surface parts only, whereas the middle parts of the reconstructed images differ significantly from the optical and X-ray images. Therefore, the practical application of the results is not clear. The authors may want to describe it in more details. For instance, they stated that photoacoustic approach was used in dentistry to identify microscopic cracks, lesions, and distribution of dental calcium for demineralization studies. How the presented results relate to this application?

Specific question: in the Ref. 27, the signals were acquired in the backward mode, whereas in the manuscript the transducer is placed in a forward direction. The authors should clearly indicate the influence of this change in measurement scheme on the reconstruction algorithm. Was the major modification of the algorithm required? If so, it should be tested on simpler objects first, like it was done in Ref. 27.

Reviewer 3 Report

Comments and Suggestions for Authors

This work has been highly anticipated. The authors possess a thorough understanding of the chemical components of teeth and Photoacoustic imaging (PAI). Their PA system and reconstruction process seems very effective. I have consistently advocated the identical PAI setup for dental specimens. This method (specially the experimental setup with rotating sample) is the most effective way to scan a tooth sample. This work is poised to garner a high volume of citations. Excellent job. Please 'Accept' it in its current form. 

Round 2

Reviewer 2 Report

Comments and Suggestions for Authors

Tha authors have improved the manuscript according to my comments.